# Metabolic Adaptations Determine the Evolutionary Trajectory of TOR Signaling in Diverse Eukaryotes

**DOI:** 10.3390/biom15091295

**Published:** 2025-09-08

**Authors:** Kyle Johnson, Dellaraam Pourkeramati, Ian Korf, Ted Powers

**Affiliations:** 1Department of Molecular and Cellular Biology, College of Biological Sciences, University of California, Davis, CA 95616, USA; kyajohnson@ucdavis.edu (K.J.); dpourkeramati@ucdavis.edu (D.P.); ifkorf@ucdavis.edu (I.K.); 2Genome Center, College of Biological Sciences, University of California Davis, Davis, CA 95616, USA

**Keywords:** TOR signaling, eukaryotic evolution, metabolism, phylogenomics, autotrophy, heterotrophy, mixotrophy

## Abstract

Eukaryotes use diverse nutrient acquisition strategies, including autotrophy, heterotrophy, mixotrophy, and symbiosis, which shape the evolution of cell regulatory networks. The Target of Rapamycin (TOR) kinase is a conserved growth regulator that in most species functions within two complexes, TORC1 and TORC2. TORC1 is broadly conserved and uniquely sensitive to rapamycin, whereas the evolutionary distribution of TORC2 is less well-defined. We built a sensitive hidden Markov model (HMM)-based pipeline to survey core TORC1 and TORC2 components across more than 800 sequenced eukaryotic genomes spanning multiple major supergroups. Both complexes are present in early-branching lineages, consistent with their presence in the last eukaryotic common ancestor, followed by multiple lineage-specific losses of TORC2 and, more rarely, TORC1. A striking pattern emerges in which TORC2 is uniformly absent from photosynthetic autotrophs derived from primary endosymbiosis and frequently lost in those derived from secondary or tertiary events. In contrast, TORC2 is consistently retained in mixotrophs, which obtain carbon from both photosynthesis and environmental uptake, and in free-living obligate heterotrophs. These findings suggest that TORC2 supports heterotrophic metabolism and is often dispensable under strict autotrophy. Our results provide a framework for the evolutionary divergence of TOR signaling and highlight metabolic and ecological pressures that shape TOR complex retention across eukaryotes.

## 1. Introduction

Within the eukaryotic tree of life, organisms have evolved different strategies for acquiring carbon and other sources of energy. Autotrophs fix inorganic carbon through photosynthesis, heterotrophs consume organic compounds derived from other organisms, and mixotrophs combine both strategies, using light energy while also assimilating external organic compounds [1,2,3]. Some organisms, including many parasites, exhibit highly specialized forms of nutrient acquisition, often relying on host-derived resources [4,5]. These strategies represent fundamental ecological and physiological adaptations that shape how cells grow, divide, and survive in different environments. In all cases, organisms depend on signaling networks that sense nutrients and modulate biosynthetic activity according to nutrient availability [6].

One essential regulator of cell growth in eukaryotes is the protein kinase TOR (for “Target of Rapamycin”), which functions within two distinct protein complexes, TORC1 and TORC2 [7,8]. TORC1 is highly conserved and controls many nutrient-regulated processes, such as protein synthesis, ribosome biogenesis, and autophagy, and is uniquely sensitive to the macrolide antibiotic and anti-cancer drug rapamycin [7,9,10,11,12]. TORC2, by contrast, is insensitive to rapamycin and remains less well characterized [13,14,15]. Initially identified in budding yeast through its role in actin cytoskeleton organization [16], TORC2 is known to regulate cell wall integrity, sphingolipid biosynthesis, and cellular responses to plasma membrane stress [7,10,17,18,19]. In mammalian cells, TORC2 modulates the actin cytoskeleton and lipid biosynthesis, contributes to insulin signaling, and supports cell survival and proliferation [8,20,21,22,23]. These diverse functions suggest that TORC2 integrates environmental and metabolic cues to modulate cellular homeostasis and metabolism, including roles in membrane remodeling, lipid synthesis, and stress-responsive signaling. TORC2 is also involved in glucose signaling in fission yeast, highlighting a role in carbon utilization [24,25].

In mammals, TORC1 is composed of TOR (also called mTOR or MTOR), the small beta-propeller protein LST8, and the large scaffolding protein RAPTOR [10,26,27,28]; TORC2 shares TOR and LST8 but includes distinct components, notably a different scaffolding subunit, RICTOR, and an additional essential protein, SIN1 [20,29,30,31] (Figure 1A). Formation of these complexes has been proposed to be mutually exclusive, with RAPTOR and RICTOR defining their respective assemblies as well as helping to recruit distinct substrates for phosphorylation by TOR [10,28,31,32]. Orthologs of these subunits are conserved within diverse eukaryotic lineages, suggesting that both complexes are evolutionarily conserved [33,34]. Nevertheless, TORC2 appears to be absent in certain lineages, most notably plants and algae within the Archaeplastida lineage [35,36]. This absence has raised questions about when TORC2 evolved and whether it was present in the earliest deep-branching eukaryotes [37]. Moreover, while rare, there are also examples of organisms that lack TORC1, including certain ciliates and phylogenetically related parasites [33,34]. Together, these differences highlight potential plasticity in the architecture of TOR signaling and underscore the importance of comprehensive phylogenetic analyses to evaluate the retention or loss of TORC1 and TORC2 and to understand what these patterns may reveal about their function.

Previous phylogenetic surveys of TOR have provided important insights into the distribution and conservation of TORC1 and TORC2 [33,34,38]. While these studies laid critical groundwork for understanding TOR evolution, they were necessarily limited by the scope of available genome data and challenges in detecting highly divergent orthologs using standard sequence similarity methods. With the expansion of available genome data and improved approaches for ortholog detection, we chose to revisit the evolutionary history of TORC1 and TORC2 at greater resolution. We anticipated that a lineage-focused comparison would clarify not only when and where these complexes were lost but also inform our understanding of the selective pressures that shaped their retention.
Figure 1Overview and strategy. (**A**) Schematic of TORC1 and TORC2. Indicated components include TORC1- (RAPTOR) and TORC2- (RICTOR and SIN1) specific proteins as well as shared proteins TOR and LST8. (**B**) Simplified diagram of an unrooted eukaryotic tree of life including LECA; depicted major clades as defined by [39]. Indicated are major proposed divisions of Diphoda and Optimoda [40]. Note that “Excavata” is included to describe a historical linkage between Metamonada and Discoba lineages [40]. Cartoon images of example species are from [41]. (**C**) Overview of bioinformatic pipeline developed for this study. Starting inputs include protein datasets and associated metadata from NCBI and JGI. Core components of TORC1 and TORC2 (Figure 1A) were identified using HMMER (version 3.4) with PANTHER HMMs. BUSCO (version 5.8.3) was used to assess proteome completeness, and BLAST (version 2.17.0) was applied to verify selected HMMER results and generate a scoring rubric. In cases of low BUSCO completeness, DIAMOND (version 2.1.12.166) was used to query SRA read data. All results were integrated into a final dataframe, which included taxonomic, metabolic, and alignment data. Phylogenetic relationships were mapped using the NCBI Common Tree, and downstream visualizations included scoring summaries and component (presence/absence) analyses (see Section 2. Materials and Methods). Color coding denotes similar inputs, outputs, and operations used: gold, publicly available data and information; green, computational tools; blue, process outputs; red, final results.
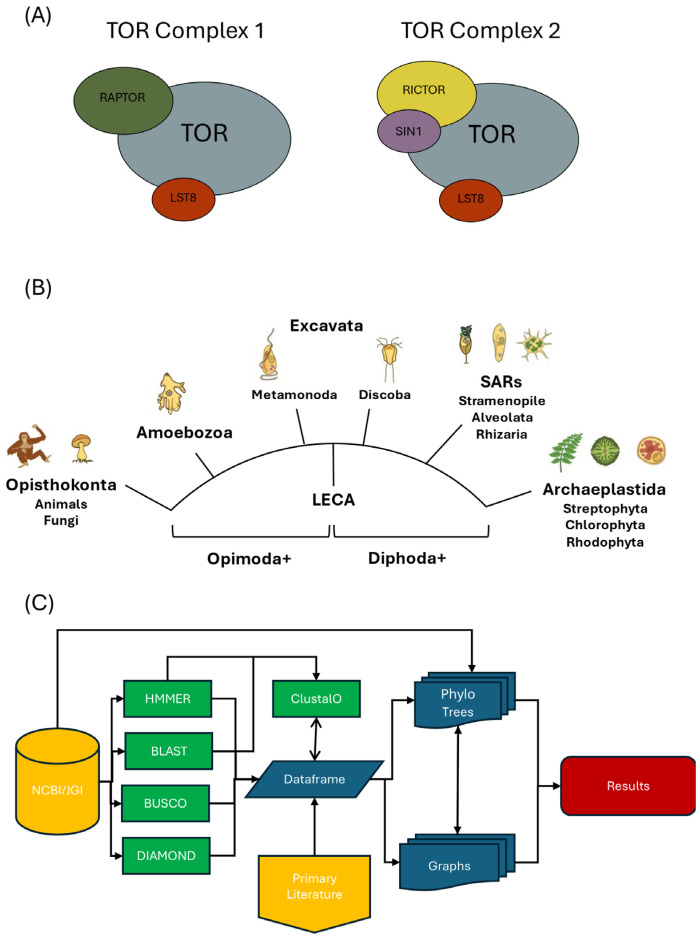



In this study, we identified core components of TORC1 and TORC2 within more than 800 sequenced eukaryotic genomes spanning major evolutionary lineages (Figure 1B). Our findings support the presence of both complexes in the last eukaryotic common ancestor (LECA) and reveal widespread, lineage-specific loss of TORC2, primarily in photosynthetic autotrophs. These results provide new insight into the functional divergence of TOR complexes and highlight how metabolic and ecological pressures have shaped their evolutionary trajectories.

## 2. Materials and Methods

### 2.1. Data Acquisition

For the six clades that were analyzed, 800 genomes were retrieved from the NCBI and JGI Phycocosm databases [42,43,44]. NCBI Sequence data was acquired through usage of “NCBI Datasets,” and JGI information was directly accessed on their website. Annotated proteomes and taxonomic information were downloaded from GenBank [45] and JGI [43]. Each species was grouped into eukaryotic clades as outlined in Burki [46].

The following HMMs where downloaded from InterPro/Panther [47,48,49]: PTHR13298 (RICTOR), PTHR12848 (RAPTOR), PTHR19842 (LST8), PTHR13335 (SIN1), and PTHR11139 (TOR). HMMs were aligned to proteomes with HMMsearch [50]. The post-processing of alignments was performed by custom BASH (version 5.1.16(1)), Python (version 3.10.13), and R (version 4.4.3) scripts.

BUSCO was used to check proteome quality and confidence of the HMMER alignments [51]. Default settings were set to “Protein” mode. Where the clade was known, the closest BUSCO database was used, otherwise the Eukaryotic database was the default.

### 2.2. Establishing a Scoring Rubric for Strength of Homology

Relational graphs were generated to determine a scoring rubric for the strength of the protein homology based upon HMMER alignments (Appendix A). Based on the methods of S.R. Eddy [50] and Pearson [52], a scoring rubric was determined using the overall, one domain, and clustering of bit scores. Using BLAST [53] on random samplings of individual proteins led to the following: (i) Bit score results below 100 were not considered in the final analysis due to lack of alignment with known components. (ii) Bit scores between 100 and 150 were considered a low hit. Alignment with known components was seen with truncation or incompleteness. (iii) Bit scores between 150 and 300 were considered medium hits. Alignments with known components had truncation or incompleteness but were characterized by stronger alignment to protein-specific domains. Bit scores above 300 were considered high hits. Alignment with known components showed strong domain alignment and overall coverage as well as sequence conservation.

The maximum value of domain and overall bit scores were then selected to indicate the presence of proteins within each organism. To note, however, this selection could not establish if multiple copies of proteins existed within the same organism. For instance, it is known that *A. thaliana* has two RAPTOR proteins [54]. Future iterations of the scoring rubric would attempt to resolve this issue and enable detection of protein copies and variants.

To confirm overall accuracy, all proteins found in the HMMER alignment were aligned with Clustal Omega (version 1.2.4) [55] to create multiple sequence alignments (MSAs). Clustal Omega was run on program default settings. Three control species were used as references for protein domain regions, as confirmed in Pfam and InterPro [47,48,49]: *H. sapiens*, *S. cerevisiae*, and *S. pombe*. JALVIEW [56] was used for the editing and visualization of MSAs.

Additional verification was performed for Streptophyta and Chlorophyta species. Certain aligned proteins had unexpected bit scores and alignments for TOR complex components, inconsistent with their predicted phylogeny. These proteins were additionally analyzed using BLAST and Multiple Sequence Alignments (MSAs). The results indicated that these proteins were often contaminates (e.g., fungi) or failed to have any aligned domain sections, indicating improper annotation. As such, these proteins were not analyzed further.

### 2.3. Sequence Read Archive (SRA) Data Analysis

For proteomes with low completeness scores, a high fragmentation score, as phylogenetically close to a species with a given component, and/or other indicative patterns of missing components, SRA reads, if available, were obtained from NCBI [42] and then analyzed with Diamond [57] using a library of currently known components. To test the accuracy of a Diamond search, several negative controls from multiple super groups were conducted on species highly inferred to lack at least one protein component, as seen in the HMMER alignment.

SRA species ID, run ID, and average length were decoded and collected using NCBI Taxonomic ID and custom Python scripts. The reads were required to be single stranded, transcriptomic, RNA-Seq data produced by Illumina sequencers. The average short-read length desired was approximately 300 nts, but shorter or longer reads (excluding long PacBio reads) were searched if the desired length was unavailable.

To determine the presence of proteins within read data, the following general criteria were used on Diamond’s outputs: a threshold E-value of E-10, bit scores greater than 50 [52], and alignments corresponding to known protein domain regions. In situations where individual short-read matches were shorter than 30 nucleotides, the E-value and bit score became unreliable measures. To compensate, it was expected that the percentage similarity score would be ~70% or higher.

In instances of poor genome assembly, reads may be shorter than expected and/or only partially sequenced. For instance, the Rhizarian clade was found to be highly fragmented and incomplete. Consequently, validity of the presence of matches had to be determined in the context of the organism.

### 2.4. Generating Phylogenetic Trees

Phylip Trees were retrieved from NCBI Common Tree [44,45] using NCBI Organism Taxonomic IDs as a reference. Tree visualizations were generated using R language and the ggtree library [58]. Tree tip labels were colored based on the level of genome completeness as established by BUSCO analysis. The primary heatmap was generated using HMMER alignment scores. The secondary heatmap was generated using metabolic information collected through the primary literature review. Trees were further divided into individual eukaryotic clades [46].

## 3. Results

### 3.1. Phylogenetically Informed Detection of TOR Complex Components

We developed a bioinformatic pipeline to identify TOR complex components in a broad set of eukaryotic genomes (Figure 1C and Materials and Methods). Our dataset included over 800 sequenced genomes spanning several major groups: Archaeplastida, SAR (Stramenopiles, Alveolates, and Rhizarians), and two separate deep-branching clades, Discoba and Metamonada, that until recently had been categorized together within the supergroup Excavata [39,40,46,59] (for historical purposes, we include reference to this nomenclature here). We searched each genome for the five core TOR complex proteins—TOR, RAPTOR, RICTOR, SIN1, and LST8—using a hidden Markov model (HMM)-based approach termed HMMER for homology detection [50]. Positive hits were assigned to one of four confidence tiers: high, medium, low, or absent (Materials and Methods) (Appendix A). To support protein identity and help resolve ambiguous cases, we constructed multiple sequence alignments (MSAs) to confirm the conservation of key architectural domains (Appendix A). Proteome completeness was independently assessed using the Benchmarking Universal Single-Copy Orthologs (BUSCO) tool [51], and results were compiled into a searchable database of TOR component homologs within the dataset (Appendix A).

To provide evolutionary context and detect broader patterns of conservation, we constructed a phylogenetic tree based on NCBI taxonomy and organized it according to an updated supergroup framework [46], mapping each identified TOR component within the tree (Figure 1C). Importantly, the topology of this tree was entirely independent of the distribution of our TOR component data itself. Thus, clusters of conserved or absent components within monophyletic groups provided strong internal validation of our pipeline. These lineage-level patterns also helped distinguish true evolutionary losses from possible false negatives, for example, due to incomplete genome availability (e.g., a low BUSCO score). In instances where a missing component appeared as an outlier because it was present in closely related species, we applied additional methods for detection (Figure 1C and Materials and Methods), refinements that improved our ability to identify orthologs.

### 3.2. Lineage-Specific Variation in the Distribution of TOR Complexes

Our analysis confirms results of prior studies concluding that TOR, LST8, and RAPTOR are conserved within all major eukaryotic lineages, supporting the central importance of TORC1 in cell signaling (Figure 2A). Sole exceptions exist within the Alveolates clade, including intracellular parasites and ciliates, as described below. By contrast, the TORC2-specific components RICTOR and SIN1 exhibit marked variation among different lineages (Figure 2A). We observe that RICTOR is more conserved than SIN1, with the latter never found in the absence of RICTOR. These differences are consistent with the proposed essential role for RICTOR in TORC2 assembly, whereas SIN1 may serve distinct regulatory roles within lineages [17,20,31]. We attribute prior conclusions that SIN1 can be found in the absence of RICTOR to differences in methodologies [33].

We find that RAPTOR and/or RICTOR are always accompanied by the presence of both TOR and LST8. Accordingly, in our analysis below, we equate the presence of RICTOR, with or without SIN1, with the presence of TORC2 in a species. Similarly, when RAPTOR is present, we assume the presence of TORC1. At present, we cannot exclude the possibility that these complexes exist using non-canonical, presently uncharacterized, species-specific components. Moreover, in species lacking both RAPTOR and RICTOR, within intracellular parasites within the Alveolata supergroup, it is possible that TOR, with or without LST8, retains residual or novel signaling functions outside canonical TOR complexes. We attribute prior conclusions that TORC1 or TORC2 can exist without LST8 to differences in methodologies [36].

Importantly, we observe that TORC1 and TORC2 are both present in the two deep-branching lineages included in our analysis, Discoba and Metamonada (Figure 2A,B and Appendix A). These early-diverging groups, historically categorized together as “Excavata”, are of particular interest for reconstructing the ancestry of eukaryotes because of their phylogenetic placement near the base of the eukaryotic tree. In particular, Discoba and Metamonada have been proposed to fall on opposite sides of the deepest eukaryotic split, corresponding to what is termed the Diphoda+ and Opimoda+ lineages, respectively [39,40,46,59] (Figure 1B). All examined species in these clades with strong BUSCO scores possess RAPTOR and RICTOR, and a small subset also contain identifiable homologs of SIN1. Although SIN1 is less frequently detected, its presence together with the presence of RICTOR supports the conclusion that a canonical TORC2 was present in the last eukaryotic common ancestor (LECA) and was subsequently lost during the evolution of specific lineages.

A striking finding is the complete absence of TORC2 in all sampled members of Archaeplastida, a monophyletic group that includes Streptophyta (land plants and related green algae), Chlorophyta (a separate clade of green algae), and Rhodophyta (red algae) (Figure 2A,B and Appendix A). Loss of TORC2 in land plants and green algae has been previously noted, but the breadth and consistency of this pattern within more than 450 genomes, together with its presence in both Discoba and Metamonada, supports a model in which TORC2 was lost in the ancestor to Archaeplastida, prior to the subsequent diversification of this lineage [60].

Within the SAR supergroup, we observe considerable variation in the presence of TORC1 and/or TORC2. Thus, while many species retain both complexes, others maintain only one or lack identifiable components altogether (Figure 2A,B). Within these supergroups there appear to have been multiple independent loss events with respect to TORC1 and/or TORC2. This diversity within SAR taxa highlights the evolutionary plasticity of TOR signaling and suggests that TOR complex retention may be shaped by distinct ecological pressures and metabolic requirements. To explore these relationships, we examined TOR architecture in each SAR lineage in greater detail. Because our initial analysis suggested a correlation between TOR complex composition and primary metabolic strategies used by individual species, we assigned each species a dominant metabolic strategy (e.g., autotrophy, mixotrophy, and heterotrophy, as well as different forms of symbiosis), based on the published literature and curated databases, and mapped these classifications onto our phylogenetic framework.

### 3.3. Tracing TOR Complex Differences in SAR Lineages

#### 3.3.1. Stramenopiles

Stramenopiles (also known as heterokonts) are a diverse group of protists that include a photosynthetic Ochrophyta clade, which acquired its plastid through a secondary endosymbiosis with a red alga, as well as several non-photosynthetic lineages [61,62] (Figure 3 and Appendix A). Ochrophytes encompass autotrophic groups such as Bacillariophyta (e.g., diatoms), Phaeophyceae (e.g., brown algae), and Pelagophyceae, along with mixotrophic species within the Parmales lineage (e.g., Triparma). Our data reveal two independent lineage-specific losses of TORC2 within Ochrophyta: TORC2 is uniformly absent in all Bacillariophyta and Pelagophyceae species examined, whereas it is retained in Phaeophyceae species (e.g., Ectocarpus), all Parmales species, and in closely related autotrophic and mixotrophic taxa (e.g., Nannochloropsis) (Figure 3 and Appendix A).

In contrast, all non-photosynthetic stramenopiles in our dataset retain TORC2, with both RICTOR and SIN1 detected within species from Oomycota (e.g., Phytophthora, Peronospora, and Aphanomyces) and Hyphochytrium (e.g., pseudofungi); the only deviation is observed within the Bigyra lineage, which includes several Blastocystis species and members of the Labyrinthulomycetes, where RICTOR is retained but SIN1 is absent (Figure 3). From these results, we conclude that TORC2 was present in the ancestor of stramenopiles, with subsequent distinct lineage-specific losses within ochrophytes. In contrast, RAPTOR is present in all stramenopiles examined, underscoring the conservation of TORC1 within this supergroup (Figure 3 and Appendix A).

#### 3.3.2. Alveolata

The Alveolata supergroup consists of two major lineages, Ciliophora, a group of heterotrophic species (e.g., ciliates), and Myzozoa, which includes several lineages that stem from an ancestor that received a red algal-derived secondary plastid, likely acquired via endosymbiosis with a stramenopile-related ancestor [61,63,64]. Myzozoa encompasses the Dinophyceae (dinoflagellates), Apicomplexa, Perkinsozoa, and Chromerida lineages, all of which are represented in our dataset (Figure 4 and Appendix A). Although this group shares a common plastid origin, many lineages are no longer photosynthetic and several are parasitic, including the Apicomplexa, which retains a non-photosynthetic plastid called the apicoplast [65].

Our analysis reveals a complex pattern of TOR complex loss within Alveolata, with at least four independent loss events inferred for each complex (Figure 4). Within Dinophyceae, TORC1 and TORC2 are retained in the autotrophic *Polarelia gracilis*, but both are lost in endosymbiotic dinoflagellates and in parasitic Amoebophyra species, indicating two independent loss events for each complex. Among apicomplexans, which include intracellular parasites such as Plasmodium and Toxoplasma, both TORC1 and TORC2 are uniformly absent. In contrast, the Perkinsozoa lineage retains both complexes, as does *Vitrella brassicaformis*, a mixotrophic chromerid. Interestingly, the closely related autotroph [66] *Chromera velia* lost TORC2 but retained TORC1, indicating an additional, independent loss of TORC2 (Figure 4). In contrast to Myzozoa, ciliates (Ciliophora) lack TORC1, based on the absence of RAPTOR in all species examined. TORC2 is retained in ciliates, where RICTOR is present in all species, including Tetrahymena and Ichthyophthirius, but SIN1 is detected only within species of Paramecium (Figure 4). Among apicomplexan parasites, we observe species that lack all TOR components as well as species that retain TOR alone or in combination with LST8. As stated above, it remains unclear whether these latter organisms retain any TOR activity or whether these components represent remnants of once-functional TOR complexes.

#### 3.3.3. Rhizaria

Rhizaria is the third major lineage within the SAR supergroup and includes free-living photosynthetic (autotrophs and mixotrophs), heterotrophic, and parasitic protists. The number of rhizarians in our dataset is limited to six species, with all retaining TORC1 and TORC2 and evidence for core components TOR, LST8, RAPTOR, and RICTOR, and a single species possessing SIN1 (Figure 4 and Appendix A). These findings suggest that the ancestral rhizarian lineage possessed both complexes, with no evidence of significant secondary loss of either complex, in contrast to stramenopiles and alveolates.

We note that several rhizarian species, such as *Bigelowiella natans* and *Lotharella oceanica*, are believed to have acquired plastids via secondary endosymbiosis of green algae and are classified as mixotrophs [64]. Thus, retention of TORC2 in these species aligns with our observations regarding retention of TORC2 in mixotrophs. In contrast, *Paulinella micropora*, the only strict autotroph in our dataset, is notable for having acquired its plastid through a relatively recent primary endosymbiosis with a cyanobacterium, independent of Archaeplastida [67]. These findings suggest that TORC2 retention in Rhizaria is shaped by both metabolic strategy and evolutionary history.

### 3.4. Symbiotic Species and TOR Complex Composition

As described above, our analysis reveals that several intracellular parasites, particularly within the apicomplexans, have lost both TORC1 and TORC2. To determine whether this pattern extends to other symbiotic lineages, we examined TOR complex composition in symbiotic species within our dataset. These include both plastid-bearing and non-plastid lineages as well as taxa that occupy primarily extracellular or intracellular environments within their hosts or, alternatively, possess a significant free-living stage outside their hosts (Appendix A).

Extracellular parasites and symbionts that lack plastids generally retain both TORC1 and TORC2. This includes Trypanosoma and Leishmania (Discoba), Giardia and Trichomonas (Metamonada), and Blastocystis (Stramenopiles). These organisms typically rely on uptake of host-derived nutrients, existing in the equivalent of an extracellular environment, and all maintain both TORC1 and TORC2. In contrast, several plastid-bearing symbionts with strictly intracellular lifestyles exhibit reduced TOR complex architecture. Apicomplexans such as Plasmodium and Toxoplasma lack both TORC1 and TORC2, as does Symbiodinium, a plastid-bearing endosymbiont within the Alveolata. In contrast, other intracellular symbionts such as Trebouxia (Chlorophyta) retain TORC1 but lack TORC2. These examples suggest that for intracellular species, particularly those with stable host associations, their environment may reduce a requirement for TORC2-mediated signaling.

## 4. Discussion

Our findings support the conclusion that TORC1 and TORC2 were both present in the last eukaryotic common ancestor (LECA) and were subsequently lost in specific lineages during evolution (summarized in Figure 5A). TORC1 remains nearly universally conserved, except in ciliates and apicomplexan parasites within Alveolata, consistent with its established role as a critical and conserved regulator of growth. A compelling pattern that emerges is the absence of TORC2 in diverse species of autotrophs, including the entire Archaeplastida supergroup and in several independently evolved lineages within Stramenopiles and Alveolata. In contrast, we identified a strong bias for the retention of TORC2 within mixotrophs and heterotrophs, revealing a likely crucial role for this complex in species that acquire carbon from the environment, either exclusively or in combination with photosynthesis. Figure 5B summarizes the correlation between metabolic strategies and the presence of TORC1 and/or TORC2 in different species. Interestingly, at the scale of the Opimoda+ versus Diphoda+ split [40], our dataset shows uniform TORC2 retention across Opimoda+ lineages, while repeated losses are confined to Diphoda+ lineages (e.g., Archaeplastida and SAR), reinforcing the view that TORC2 arose early and was subsequently lost in select Diphoda+ clades.

The widespread loss of TORC2 among photosynthetic autotrophs may reflect relaxed selective pressure on its ancestral functions. In this scenario, TORC2-dependent processes, such as membrane-associated signaling, cytoskeletal remodeling, or glucose-responsive growth regulation [17], may become dispensable under conditions where energy and carbon are reliably supplied through photosynthesis. Reducing the complexity of nutrient-sensing mechanisms may provide an advantage in stable, light-rich environments where internal photosynthetic inputs outweigh variable external nutrient cues. Supporting this idea, studies of microalgae have shown that TORC1 plays the dominant role in photosynthesis and chloroplast-derived growth control, suggesting that TORC2 may be superfluous within these lineages [36]. At the same time, it is likely that alternative pathways have replaced TORC2, given its critical roles in regulating normal cell growth. For example, studies in *S. cerevisiae* have demonstrated that TORC2 contributes to oxidative stress responses, including regulation of membrane integrity under ROS exposure [68]. Photosynthetic organisms face elevated ROS from light-driven metabolism and have evolved specialized and distinct antioxidant systems and redox signaling networks [69].

Retention of TORC2 in certain autotrophs, including Phaeophyceae (brown algae), may reflect lineage-specific physiological demands, including complex multicellularity, environmental conditions, or developmental regulation [70]. These features may impose selective pressure to maintain TORC2 even in species that rely primarily on photosynthesis. In part, this underscores both the complexity of the evolution of plastid acquisition and photosynthesis and the fact that important metabolic differences exist among autotrophs [71,72]. Alternatively, other evolutionary factors may play a role in TORC2 retention. For example, the rhizarian autotroph, *Paulinella micropora*, acquired its plastid through a primary endosymbiotic uptake of a cyanobacterium relatively recently, estimated to have occurred between 100 and 140 million years ago [67]. Thus, Paulinella may represent a transitional state prior to TORC2 loss. This view is consistent with the idea that TORC2 loss is an ongoing process occurring at multiple evolutionary scales, including in the ancestor of the Archaeplastida supergroup, between subgroups (e.g., diatoms versus brown algae within Stramenopiles), and even between closely related species (e.g., *Chromera velia* vs. *Vitrella brassicaformis* in Alveolata). Moreover, examining autotrophic lineages that retain TORC2 may offer insight into conserved features of TORC2 function, including roles typically associated with heterotrophic or mixotrophic nutrient sensing.

Symbiotic species in our dataset span a range of metabolic and cellular lifestyles, including both plastid-bearing and non-plastid lineages, and vary in whether they occupy extracellular or intracellular host environments. We observe that extracellular parasites and non-plastid symbionts generally retain both TORC1 and TORC2, whereas intracellular symbionts, including apicomplexan parasites, tend to lack one or both TOR complexes. A notable exception is the Perkinsozoa clade, where Perkinsus species, including parasites of various microorganisms and animals, retain both TORC1 and TORC2 (Figure 4B). These species have a prominent free-living stage, which may contribute to the selective pressure to maintain both complexes [73]. These patterns suggest that both lifestyle and evolutionary history influence TORC2 retention and loss (Figure 5B).

While our data support an important role for TORC2 in heterotrophic metabolism, a number of exceptions reveal the likely independent evolution of alternative regulatory strategies. For example, *Nitzschia putrida* is a diatom that lacks TORC2 and appears to have lost photosynthetic capacity and reverted to a heterotrophic lifestyle [74]. This observation indicates that prior loss of TORC2 during the autotrophic phase of this species did not prevent a shift to heterotrophy. Similarly, within the Chlorophyta and Rhodophyta, several species are identified as mixotrophic, despite an absence of TORC2. Several plant species within Streptophyta are also known to be mixotrophic, including orchids, carnivorous plants, and mistletoe [1,75]. Interestingly, also within Chlorophyta, *Prototheca wickerhamii* is proposed to have lost its photosynthetic capability and reverted to heterotrophy [76,77], also in the absence of TORC2, possibly through compensation by TORC1 activity. Therefore, at least in these known cases, heterotrophy can be regained in the absence of TORC2. Finally, *Euglena gracilis*, a well-characterized mixotroph in the Discoba lineage [78,79], lacks a detectable RICTOR homolog (Appendix A). However, its low BUSCO score suggests incomplete genome coverage, and we therefore predict that TORC2 may ultimately be identified within this species.

Our findings confirm prior observations that while TORC1 is conserved among free-living species, one notable exception is found in ciliates, which lack RAPTOR but possess TORC2. The selective loss of TORC1 in ciliates suggests a possible reconfiguration of TOR signaling in this lineage, potentially involving novel adaptor proteins or alternative regulatory strategies, bypassing the canonical role for TORC1. Investigating TOR pathway function in ciliates may help define the minimal requirements for TOR activity as well as identify novel signaling components that compensate for canonical TORC1 activity.

Our findings are, in general, consistent with a recent survey of TOR signaling in photosynthetic microalgae, including widespread loss of TORC2 in autotrophs derived from primary and secondary endosymbiosis [36]. Notably, this study also identified retention of TORC2 in Ectocarpus (brown algae) but not in closely related diatoms or pelagophytes. In addition, loss of TORC2 was reported in other autotrophs within Haptophyta, Cryptophyta, and Glaucophyta lineages, which were not included in our dataset. Some reported anomalies, such as the absence of LST8 in Cyanophora paradoxa or of all components but TOR in a Cryptophyceae species, which are free-living photosynthetic organisms, may reflect limitations in genome completeness or challenges in ortholog detection. In this regard, we have been able to detect LST8 using our HMMER-based pipeline in a recently released proteome for *Cyanophora paradoxa* (our unpublished results).

While our analysis focuses on the core components of TORC1 and TORC2, it is important to acknowledge the broader signaling context in which these complexes operate. TORC1 has been extensively characterized in many species, particularly within Opisthokonta (including humans and budding yeast), with well-defined upstream regulators, such as Rag GTPases, the TSC complex, and Rheb, as well as downstream effectors, including S6 kinase, 4E-BP, and Atg13 [7,8]. In contrast, TORC2-associated signaling partners are less well understood. Validated downstream targets include PKC, SGK, and AKT in mammals, the related AGC kinases PKC1, YPK1, and YPK2 in *S. cerevisiae*, and GAD8 in *S. pombe* [17,20]. However, conserved upstream regulators and additional downstream targets of TORC2 remain poorly defined. A comprehensive understanding of upstream inputs and downstream substrates will be essential for determining how TOR signaling networks have been remodeled within different lineages. This will be particularly important for elucidating the role of TORC2, as identifying its interacting partners and regulatory circuits may clarify the metabolic strategies that depend on its retention. Indeed, a recent study reconstructing TORC1 signaling networks in early-diverging fungi within Opisthokonta underscores the power of phylogenomics to uncover lineage-specific regulatory innovations in TOR signaling [80].

## 5. Conclusions

In summary, our findings suggest TORC2 signaling is shaped by metabolic strategy, ecological context, and lineage-specific adaptations, where it is dispensable under certain conditions, particularly in autotrophic species. Investigating how TOR signaling is maintained or reconfigured in species with divergent TOR complex architectures, as identified here, may clarify the minimal requirements for TOR function. Moreover, our findings suggest testable hypotheses, for example, that TORC2 is likely essential for the growth of mixotrophic species (e.g., *Vitrella brassicaformis*) under heterotrophic but not autotrophic growth conditions. Finally, continued improvements in genome completeness and annotation will also be essential for resolving ambiguous cases and revealing additional modes of TOR pathway evolution, including additional clades within the larger eukaryotic evolutionary tree beyond those reported here.

## Figures and Tables

**Figure 2 biomolecules-15-01295-f002:**
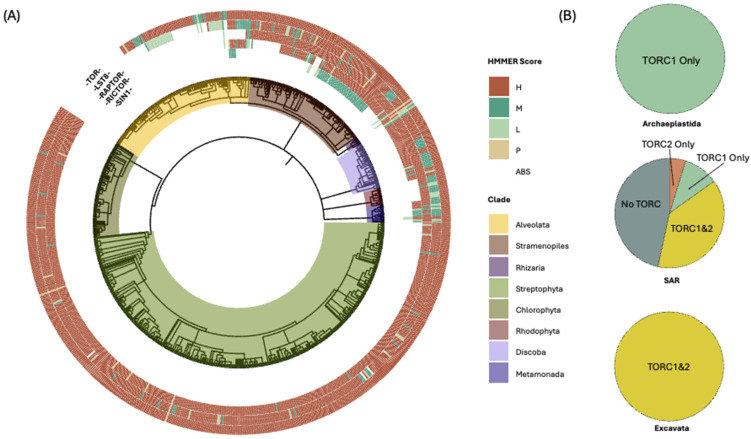
Phylogenetic survey of TOR complex components. (**A**) Circular phylogenetic tree of all surveyed species, with outer rings indicating HMMER alignment scores for the five core TOR components: TOR (outermost ring), LST8, RAPTOR, RICTOR, and SIN1 (innermost ring). Each tick mark represents a detection event, color-coded by alignment strength: high confidence (H; >300, terracotta), medium (M; 150–300, dark green), and low (L; 100–150, light green). Probable matches identified from short-read data using Diamond are shown in tan (P). Absent components are indicated in white (ABS). Clades are shaded according to their taxonomic grouping. (**B**) Charts depicting the proportion of species within three major clades, Archaeplastida, SAR, and Excavata, that possess TORC1 only (teal), TORC2 only (red), both TORC1 and TORC2 (yellow), or neither complex (gray). All surveyed species within each clade are included. While Excavates universally retain both TOR complexes, Archaeplastida uniformly lacks TORC2, and SAR exhibits the greatest variability.

**Figure 3 biomolecules-15-01295-f003:**
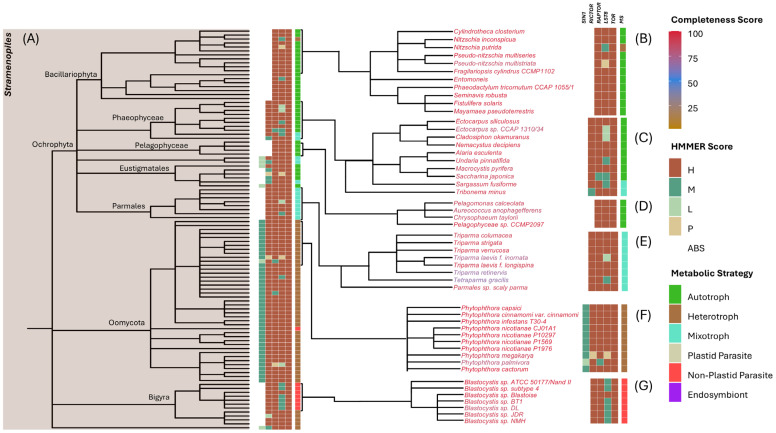
TOR complex composition and metabolic strategies within Stramenopiles. (**A**) Phylogenetic tree of curated species, with major clades within the supergroup indicated. Heatmap bars indicate detection confidence (HMMER Score) for core TOR complex components and inferred metabolic strategies, with color codes as described in the key (See Figure 2 legend for details). (**B**–**G**) Expanded views highlight specific lineages with shared evolutionary and metabolic features. Species are color-coded by BUSCO values (completeness score), as described in the key: (**B**) Bacillariophyta (diatoms), a subgroup of Ochrophyta. Includes primarily autotrophic species such as *Fragilariopsis cylindrus* and the non-photosynthetic heterotroph *Nitzschia putrida*; (**C**) Phaeophyceae (brown algae), a subgroup of Ochrophyta. Multicellular and primarily autotrophic or mixotrophic taxa, including *Ectocarpus siliculosus* and *Saccharina japonica*; (**D**) Pelagophyceae, a subgroup of Ochrophyta. Includes unicellular marine autotrophs such as *Aureococcus anophagefferens*; (**E**) Parmales, primarily mixotrophic organisms including Triparma species; (**F**) Oomycota, a heterotrophic clade that includes filamentous fungi-like species such as *Phytophthora infestans*; (**G**) Bigyra, a clade consisting of diverse free-living and parasitic taxa including Blastocystis species.

**Figure 4 biomolecules-15-01295-f004:**
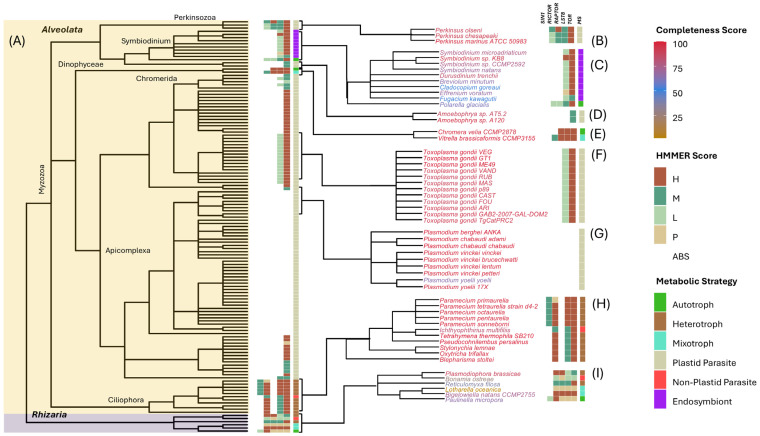
TOR complex composition and metabolic strategies within Alveolata and Rhizaria. (**A**) Phylogenetic tree of curated species, with major clades within the supergroup indicated. Heatmap bars indicate detection confidence (HMMER score) for core TOR complex components and inferred metabolic strategies, with color codes as described in the key. (**B**–**I**) Expanded views highlight specific lineages with shared evolutionary and metabolic features. Species are color-coded by BUSCO values (Completeness Score), as described in the key: (**B**) Perkinsozoa, plastid-containing facultative parasites, including Perkinsus species; (**C**) Dinophyceae, including Symbiodinium species and the free-living autotroph *Polarella glacialis*; (**D**) Dinophyceae, including Amoebophyra, a group of parasitic dinoflagellates; (**E**) Chromerida, including *Chromera velia* and *Vitrella brassicaformis*, close relatives of Apicomplexans; (**F**) Apicomplexans (e.g., Toxoplasma gondii), intracellular parasites retaining the non-photosynthetic apicoplast; (**G**) Plasmodium species, a subgroup of Apicomplexans; (**H**) Ciliates, a plastid-lacking clade of heterotrophs and parasites; (**I**) Rhizaria, including primarily heterotrophic and mixotrophic taxa, as well as the photosynthetic species *Paulinella micropora*.

**Figure 5 biomolecules-15-01295-f005:**
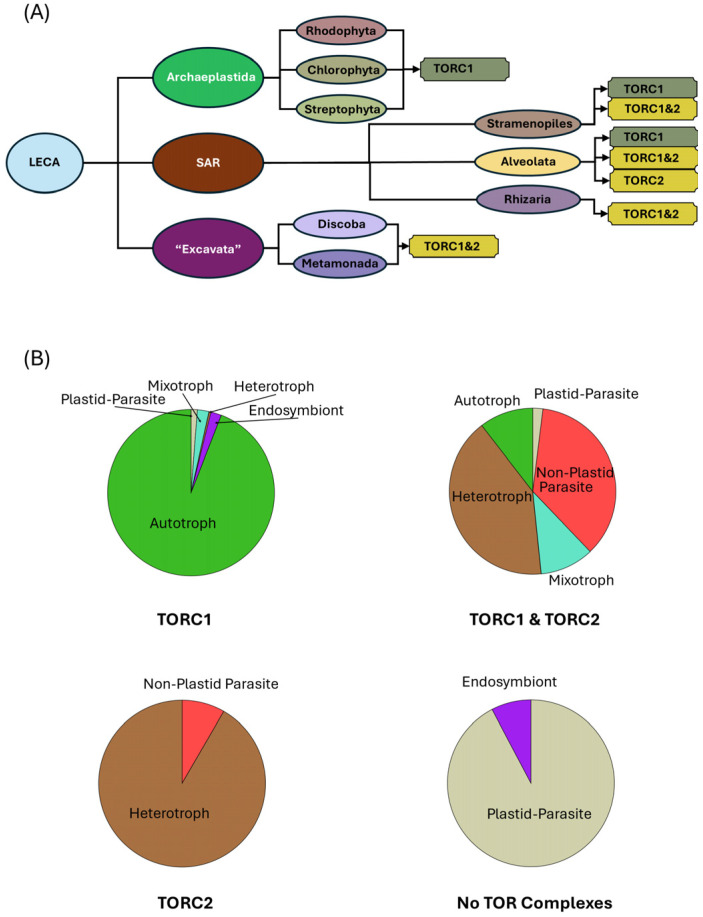
Evolutionary distribution of TOR complexes and associated metabolic strategies. (**A**) Summary illustrating inferred retention or loss of TORC1 and TORC2 within major eukaryotic lineages. See text for details. (**B**) Charts summarizing major metabolic strategies of species grouped by TOR complex composition. Species retaining only TORC1 are predominantly autotrophic, with smaller contributions from mixotrophic, heterotrophic, and endosymbiotic taxa. Species lacking both complexes are primarily plastid-containing parasites or endosymbionts. Species retaining only TORC2 (represented here by ciliates) are heterotrophic or parasitic. Species retaining both TORC1 and TORC2 span a range of metabolic strategies, primarily heterotrophic and mixotrophic, in addition to non-plastid parasitic species.

## Data Availability

A full collection of scripts, HeatMaps, Trees, raw HMMER results data, combined results tables, trophic strategy tables, figures, and list of the BUSCO databases used are accessible on GitHub (https://github.com/Ted-Powers-Lab/TOR_phylogenetics, created on 3 September 2025). Further information regarding a complete collection of all R libraries used in the creation of this analysis can be found in the README file found on the GitHub repository.

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
