# Peer review of "Metabolic Adaptations Determine the Evolutionary Trajectory of TOR Signaling in Diverse Eukaryotes"

_biomolecules, 2025, doi:10.3390/biom15091295_

Round 1
Reviewer 1 Report
Comments and Suggestions for Authors
In this study, Johnson et al. took a comprehensive approach to systematically survey the available genome data to delve into the evolutionary conservation of TORC1 and TORC2 components. They have proposed a persuasive argument about the retention and loss of these signaling complexes during the evolutionary divergence of eukaryotic species. The dataset from this study, shared on GitHub, will be very useful to the TOR research community.
One trivial suggestion. A broader readership would appreciate it if “HMM” were spelled out in the Abstract.
Author Response
Reviewer Comment: One trivial suggestion. A broader readership would appreciate it if “HMM” were spelled out in the Abstract.
I have modified the Abstract as requested.
Reviewer 2 Report
Comments and Suggestions for Authors
In this article, the authors use sequence data from 800 eukaryotic genomes to analyze the evolution and function of TOR complexes. As a cell biologist who has studied TOR signaling, this was an interesting and informative analysis. For me, the most interesting discovery is that TORC2 is lost in autotrophs, which suggests that TORC2 at least initially evolved to manage how cells obtain nutrients from the environment. This fits well with our work showing that TORC2 plays a critical role in managing growth in different carbon sources in budding yeast, and it makes me wonder whether the primary function of TORC2 in all eukaryotic cells is to manage utilization of external nutrients. My only suggestion would be to put greater emphasis on key conclusions, and to feel free to speculate more about the function and evolution of TORC2. The article is well-written and I do not have any suggestions for substantial edits.
Author Response
Reviewer Comment: My only suggestion would be to put greater emphasis on key conclusions, and to feel free to speculate more about the function and evolution of TORC2. The article is well-written and I do not have any suggestions for substantial edits.
We are pleased that the reviewer liked the paper and that our conclusions are well supported by our analysis. We have speculated that the role of TORC2 in heterotrophy is connected to it role in carbon acquisition (e.g. glucose signaling), but also membrane trafficking and stress responses. At this point, we are comfortable with this level of speculation and that we have pointed out avenues for further investigation.
Reviewer 3 Report
Comments and Suggestions for Authors
In study, Johnson and colleagues use a sophisticated bioinformatics, HMM-based pipeline to detect the presence of the highly conserved eukaryotic TOR complexes (i.e. TORC1 and TORC2) across more than 800 sequenced eukaryotic genomes including all major clades. Notably, both complexes probe different intracellular and extracellular cues and transmit information to specific, non-overlapping effectors to coordinate growth and metabolism.
A major finding of the presented analysis is that both TOR complexes were present in the last eukaryotic common ancestor (LECA). Strikingly, as this study elegantly highlights, certain forms of live (e.g., photosynthetic autotrophs derived from primary endosymbiosis), have lost TORC2, while others (e.g., intracellular parasites within the apicomplexans) have even lost both TORC1 and TORC2 during evolution. These and additional observations allowed the authors to couple metabolic strategies with the presence of TORC1 and/or TORC2 and thus to delineate evolutionary pressures that may have shaped TOR complex loss and retention across eukaryotes. Overall, this is an exquisitely well-curated manuscript that timely reports on an aspect in TOR biology that has so far received little attention. The approaches and findings are highly original, and the conclusions are well justified by the presented data. I am sure that this manuscript will be met with great interest in the TOR field.
I have just a couple of minor remarks:
- Throughout the text, species names are often in regular letters, rather than in italics (see for instance S. cerevisiae and S. pombe [line 143], or Vitrella brassicaformis [line 334]; and others). I suggest checking the entire manuscript for coherence regarding this issue.
- The sequence of the supplementary figures should perhaps be reviewed, because Fig. S3 (line 249) is followed by Fig. S6 (line 263), and then Fig. S4 (line286).
- When discussing the loss of TORC2, the authors speculate that it may just have become dispensable under conditions where carbon and energy are reliably supplied. Another possibility is that TORC1 may have taken over some of TORC2’s function. This may perhaps also be the case for the Nitzschia purtrida diatom, which has likely lost TORC2 during an autotrophic phase and then became heterotrophic again?
Author Response
We are very happy that the Reviewer liked the manuscript and believes it will be a valuable addition to the field. Below are our responses to the minor comments provided.
1. Throughout the text, species names are often in regular letters, rather than in italics (see for instance S. cerevisiae and S. pombe [line 143], or Vitrella brassicaformis [line 334]; and others). I suggest checking the entire manuscript for coherence regarding this issue.
We thank the reviewer for pointing out our need to further edit the manuscript for proper use of species names and have made the requested changes.
2. The sequence of the supplementary figures should perhaps be reviewed, because Fig. S3 (line 249) is followed by Fig. S6 (line 263), and then Fig. S4 (line286).
We thank the reviewer for this suggestion. We have reordered the supplemental figures so that they are listed in numerical order in the text.
3. When discussing the loss of TORC2, the authors speculate that it may just have become dispensable under conditions where carbon and energy are reliably supplied. Another possibility is that TORC1 may have taken over some of TORC2’s function. This may perhaps also be the case for the Nitzschia purtrida diatom, which has likely lost TORC2 during an autotrophic phase and then became heterotrophic again?
We thank the reviewer for this insightful comment. When we describe species that have lossed TORC2, in some cases we have specifically indicated the need for alternative pathways to compensate for known roles of TORC2, for example in ROS mediation in plants. However, it is true that we imply there was a loss of a requirement for TORC2 in many species. We have modified the text to indicate the possibility that some TORC2 activities could have been taken over by TORC1 (e.g. line 389 in the revised manuscript: Therefore, at least in these known cases, heterotrophy can be regained in the absence of TORC2, possibly through compensation by TORC1 activity."